# NaRCan: Natural Refined Canonical Image with Integration of Diffusion Prior for Video Editing

**Ting-Hsuan Chen    Jiewen Chan    Hau-Shiang Shiu**
**Shih-Han Yen    Chang-Han Yeh    Yu-Lun Liu**
National Yang Ming Chiao Tung University

## Abstract

We propose a video editing framework, NaRCan, which integrates a hybrid deformation field and diffusion prior to generate high-quality natural canonical images to represent the input video. Our approach utilizes homography to model global motion and employs multi-layer perceptrons (MLPs) to capture local residual deformations, enhancing the model's ability to handle complex video dynamics. By introducing a diffusion prior from the early stages of training, our model ensures that the generated images retain a high-quality natural appearance, making the produced canonical images suitable for various downstream tasks in video editing, a capability not achieved by current canonical-based methods. Furthermore, we incorporate low-rank adaptation (LoRA) fine-tuning and introduce a noise and diffusion prior update scheduling technique that accelerates the training process by 14 times. Extensive experimental results show that our method outperforms existing approaches in various video editing tasks and produces coherent and high-quality edited video sequences. Code and video results are available at koi953215.github.io/NaRCan_page.

## 1   Introduction

Video editing has always been a fascinating research area. For example, style transfer transforms the original video into a completely new style, enriching the viewing experience. Other tasks include dynamic segmentation and handwriting, which all demonstrate the broad application value of video editing across various fields. Currently, diffusion model technology is becoming increasingly mature and is known for its powerful generative capabilities and frequent use in video editing. However, in video-to-video tasks, maintaining temporal consistency presents a significant challenge, particularly when applying image-based diffusion models to video editing tasks. When these models, originally designed for image generation, are applied frame-by-frame to videos, they often produce temporally inconsistent results due to their frame-independent processing nature. This limitation has motivated numerous research efforts to enhance temporal coherence through various techniques, such as optical flow guidance, latent space alignment, and cross-attention mechanisms, aiming to produce high-quality video sequences while preserving temporal consistency across frames.

Yet, even with solutions addressing temporal consistency, diffusion-based methods [16, 4, 19, 8] still encounter significant challenges in precise localized editing tasks. This is where canonical-based methods demonstrate their unique advantages. By consolidating video content into a single representative image, canonical-based methods [3, 1, 5, 45] enable intuitive and precise spatial control over editing operations while inherently maintaining temporal consistency. This unified representation allows direct application of image-based editing techniques while ensuring that modifications propagate coherently throughout the video sequence, making these methods particularly effective for a wide range of video editing tasks.

38th Conference on Neural Information Processing Systems (NeurIPS 2024).

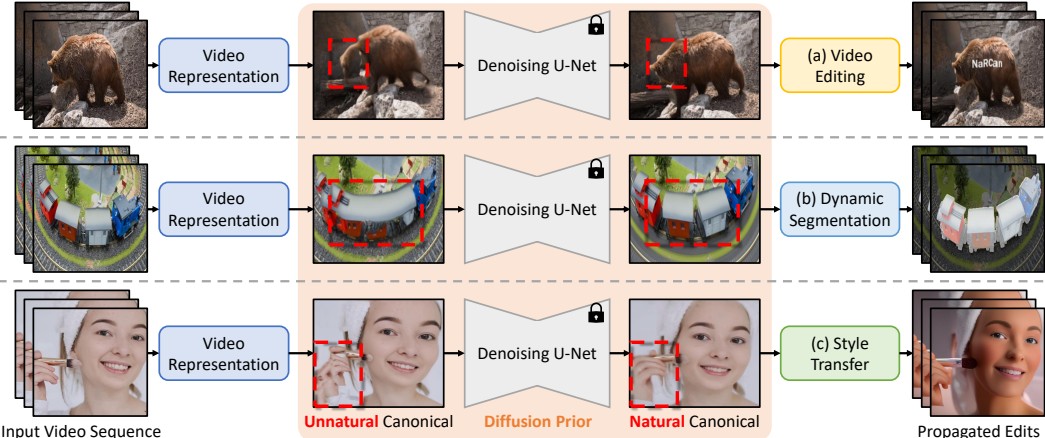

Input Video Sequence     **Unnatural** Canonical    **Diffusion Prior**    **Natural** Canonical     Propagated Edits

Figure 1: **Video representation with diffusion prior.** Given an RGB video, we can represent the video using a canonical image. However, the canonical image and reconstruction training process focuses only on reconstruction quality and could produce an unnatural canonical image. This could cause problems with downstream tasks such as prompt-based video editing. In the bottom example, if the hand is distorted in the canonical image, the image editor, such as ControlNet [75], may not recognize it and could introduce an irrelevant object instead. In this paper, we propose introducing the *diffusion prior* from a LoRA [18] fine-tuned diffusion model to the training pipeline and constraining the canonical image to be natural. Our method facilitates several downstream tasks, such as (a) video editing, (b) dynamic segmentation, and (c) video style transfer.

While previous canonical-based approaches have established the fundamental utility of canonical images in video editing, our observations suggest that enhancing the naturalness of these representations can significantly improve their effectiveness in various editing scenarios. We observe that existing methods primarily focus on reconstruction quality without explicit constraints to ensure natural canonical image generation. To address this opportunity for enhancement, we propose NaRCan, a novel hybrid deformation field network architecture that incorporates diffusion priors (Figure 1) into the training pipeline (Figure 2). This integration enables our method to generate high-fidelity natural canonical images across diverse scenarios while maintaining temporal consistency. Through comprehensive experimental evaluation, we demonstrate that NaRCan achieves superior performance compared to state-of-the-art video editing methods in both qualitative and quantitative metrics.

Our main contributions are three-fold:

- We have designed a novel deformation field structure to represent object variations throughout an entire scene. Compared to other canonical-based models, our model demonstrates superior expressive capability and faster convergence speed.

- We have effectively integrated the diffusion prior into our pipeline, enabling our method to generate high-quality natural canonical images in any scenario. Additionally, we designed a dynamic scheduling method that significantly accelerates the entire training process.

- We thoroughly evaluate our method to show the state-of-the-art video editing performance.

## 2 Related Work

**Implicit neural representation.**    Implicit neural representation [43] using coordinate-based MLP is an outstanding way to represent a video, capable of obtaining a canonical image to represent the entire video [45]. Recent methods employ hash grid encoding [42] or positional encoding [40] combined with MLP. The approach in [5] is more effective at handling spatial information, but the resulting canonical image exhibits severe distortion and warping with in-the-wild videos [11]. Therefore, we propose a hybrid deformation field method composed of homography [9] and residual deformation MLP. This model design fits the deformation information in videos better than existing methods.

**Consistent video editing.**    There are generally three approaches to video editing: (1) propagation-based, (2) layered representation-based, and (3) canonical-based. The first approach, propagation-

based, focuses on propagating information across frames [35, 20, 22, 23, 57, 63, 67]. This method can easily produce inaccurate results due to occlusions and error propagation. The second approach, layered representation-based, separates a video into foreground and background, obtaining canonical images for both [36, 25, 73, 34, 37, 38]. We can edit the entire video by editing these canonical images and then synthesizing the video afterward. However, this method heavily relies on masks. If the mask-RCNN captures incorrect targets in the preprocessing stage, the fitted foreground can be incorrect, especially in scenes with large camera movement.

The third approach, canonical-based, typically uses MLP to obtain the deformation information of each pixel to form a canonical image [45]. Transferring the video to canonical space maintains temporal consistency while editing and supports various downstream tasks, such as super-resolution and segmentation. CoDeF [45] uses this approach, but canonical images can deform severely in videos with significant camera or object movement. CoDeF suggests that using a group model could resolve these issues. However, using group CoDeF requires masks for training data obtained from SAM-track [12]. Incorrect masks for training data can result in an unnatural canonical image of the video. Even with correct masks for the foreground object, the canonical image might still be corrupted, rendering these images ineffective for video editing.

**Video processing via generative models.** Some works utilize GAN inversion [65, 78, 60, 74, 48, 26, 69] to edit images or videos. Today, numerous generative models exist for editing images. Some methods, such as GLIDE [44], DALL-E [55, 54], stable diffusion [56], and Imagen [58], are trained on millions of images, resulting in incredible generative abilities. Other methods like SDEdit [39], ControlNet [75], and LDM [2] use conditions to achieve better editing results. Instruction-based video editing methods like InstructPix2Pix [3] and another work [10] often yield sub-optimal results for different editing operations. Techniques like LoRA [18] can assist in fine-tuning to find better weights for editing. Additionally, many zero-shot diffusion-based methods [72, 16, 76, 4, 8, 19, 27] do not require model training but still need constraints to maintain temporal consistency. To address temporal consistency concerns, most methods like Tune-A-Video [70], Text2Video-Zero [27], FateZero [52], and Vid2Vid-Zero [66] incorporate cross-attention mechanisms. Some works propose training for video editing, such as Imagen Video [17] and Make-A-Video [59], but these require large datasets and significant computational resources. Unlike these methods, MeDM [13] uses a flow-coding algorithm to solve this problem. Canonical-based design does not require another mechanism to maintain temporal consistency, however. Once the canonical image is edited, the changes can be propagated to every frame using a deformation field.

**Lifting the naturalness of canonical image by diffusion models.** The diffusion prior has been applied in various domains. Reconfusion [71] is a few-shot novel view synthesis work that introduces a diffusion model before optimizing sampled novel views. Dreamfusion [50], a text-to-3D work, introduces score distillation sampling (SDS) loss, referencing a 2D diffusion model to optimize 3D outputs. This approach inspires us to utilize the diffusion model to improve performance. Other methods like [6, 7, 21, 41, 33, 61, 64, 68, 79] also employ diffusion prior for text-to-3D tasks.

Several diffusion models focus on text-to-image generation [53, 56, 58, 62]. Additionally, some diffusion models can refine corrupted images to make them appear more natural. We propose adding a diffusion model to our pipeline (Figure 2) to enhance the naturalness of our canonical images. Our goal is to improve the restoration of canonical images using the diffusion model. While we create canonical images using a hybrid deformation field, this method might not always deliver optimal performance, especially in scenarios with dramatic motion changes or severe non-rigid transformations. The capabilities of the hybrid deformation field are still limited in such cases. Therefore, we aim to introduce a diffusion model to make our canonical images more natural, enhancing the effectiveness of video editing. This research has significant practical implications as it can improve the quality and realism of video editing, benefiting various industries such as film production, advertising, and virtual reality.

## 3 Method

In this section, we first introduce our hybrid deformation modeling by combining homography and deformation MLP in Section 3.1. Subsequently, we elaborate on how we integrate the diffusion prior from a LoRA fine-tuned latent diffusion model to ensure the naturalness of our canonical image representation in Section 3.2. Finally, we provide an additional way to improve the quality of our

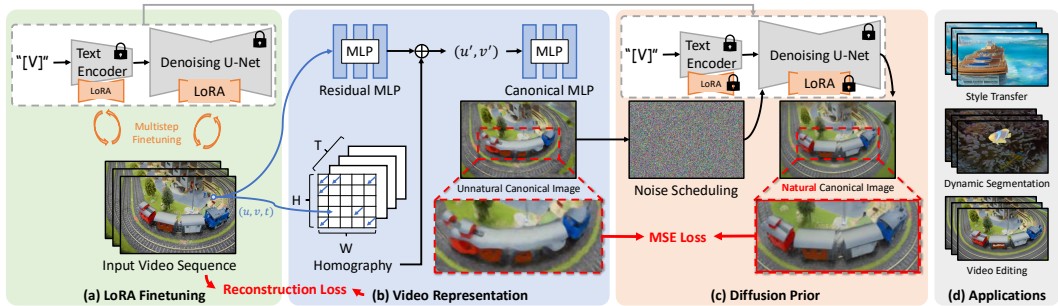

Figure 2: **Our proposed framework.** Given an input video sequence, our method aims to represent the video with a *natural* canonical image, which is a crucial representation for versatile downstream applications. (a) First, we fine-tune the LoRA weights of a pre-trained latent diffusion model on the input frames. (b) Second, we represent the video using a canonical MLP and a deformation field, which consists of homography estimation and residual deformation MLP for non-rigid residual deformations. By relying entirely on the reconstruction loss, the canonical MLP often fails to represent a natural canonical image, causing problems for downstream applications. *E.g.*, image-to-image translation methods such as ControlNet [75] may not be able to recognize that there is a train in the canonical image. (c) Therefore, we leverage the fine-tuned latent diffusion model to regularize and correct the unnatural canonical image into a natural one. Specifically, we sophistically design a noise scheduling corresponding to the frame reconstruction process. (d) The natural and artifacts-free canonical image can then be facilitated to various downstream tasks such as video style transfer, dynamic segmentation, and editing, such as adding handwritten characters of "NaRCan".

video representation by separating multiple canonical images and describing the necessary changes for downstream tasks in Section 3.3. Figure 2 shows our proposed framework.

## 3.1 Hybrid Deformation Field for Deformation Modeling

Traditional methods often rely on direct predictions of $\Delta u$ and $\Delta v$, which means the displacement of pixel points $u$, $v$ at time t, by using an MLP $g(\cdot, \cdot, \cdot)$ and query the RGB color by another canonical image MLP $f(\cdot, \cdot)$:

$$\Delta u, \ \Delta v = g(u, \ v, \ t), \quad [R, \ G, \ B] = f(u + \Delta u, \ v + \Delta v), \tag{1}$$

supplemented with a TVFlow regularization term to prevent overfitting by constraining the magnitude of spatial deformations. While this regularization effectively prevents extreme deformations, it merely imposes magnitude constraints without providing meaningful guidance for modeling complex spatial transformations. To address this limitation, we propose a hybrid deformation field architecture composed of a trainable homography matrix $H(u, v, t)$ and residual deformation MLP:

$$u', \ v' = H(u, \ v, \ t) + g(u, \ v, \ t), \quad [R, \ G, \ B] = f(u', \ v'). \tag{2}$$

Unlike the simple constraint-based approach of TVFlow, our method leverages homography to provide global displacement information as structural guidance for the subsequent residual deformation MLP. This hierarchical design enables the residual deformation MLP to learn and express the deformation field more accurately and effectively by building upon the initial geometric transformation provided by the homography matrix.

## 3.2 Diffusion Prior

The canonical image encompasses all information within the entire video and can be reconstructed for each original video frame using the deformation field outlined in Section 3.1. Thus, editing solely the canonical image yields a temporally coherent edited video. However, editing tasks such as drawing or writing on objects or editing based on ControlNet [75] require a natural image as input to produce meaningful edited images. Existing canonical base methods do not incorporate mechanisms to ensure that the generated image is natural; instead, they rely solely on the model's ability to learn a natural image. However, when encountering scenarios with camera movement or significant changes in video content, these existing techniques cannot adapt to such drastic variations. The model may generate a canonical image that is nearly impossible to edit (rendering it devoid of any subsequent value).

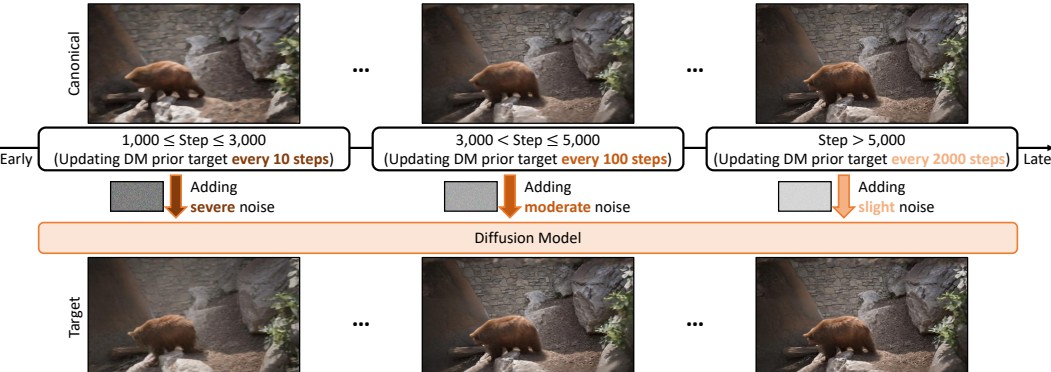

Figure 3: **Noise and diffusion prior update scheduling.** Initially, our model fits object outlines before the fields converge and without the diffusion prior, resulting in unnatural elements in the canonical image due to complex non-rigid objects. Upon introducing the diffusion prior with increased noise and update frequency, the model learns to generate natural, high-quality images, leading to convergence. Thus, the strength of noise and the update frequency will also decrease. Moreover, it's worth mentioning that update scheduling cuts training time from 4.8 hours to 20 minutes.

To address these challenges, we introduce diffusion priors, which successfully mitigate this issue. Our method can generate high-quality canonical images through diffusion priors, providing valuable inputs for various video editing tasks.

**LoRA fine-tuning.**    To enhance the current diffusion model's ability to represent all video content better, we introduced a special token specific to this scene. We then fine-tuned the LoRA weight of the pre-trained diffusion model. This ensures that the diffusion model generates high-quality natural canonical images tailored to the testing sequence rather than randomly generating natural images that do not belong to the scene.

**Noise and diffusion prior update scheduling.**    While the hybrid deformation field technique mentioned in Section 3.1 already produces better canonical images than other existing methods, it still needs to improve. As Figure 9(b) depicts, canonical images generated solely relying on homography and residual deformation MLP still exhibit various degrees of distortion and unnatural characteristics. Therefore, integrating diffusion priors becomes imperative. We extract the canonical region currently observed by the model and calculate diffusion loss with the target image generated by the diffusion model to ensure the generation of natural canonical images. However, generating a target image at each step would significantly prolong the training phase. Hence, we propose a hierarchical update scheduling to accelerate the process, which is shown in Figure 3. In the initial stages of training, when the deformation field has not yet converged, more substantial noise is introduced to allow the diffusion model to dominate the scene of the canonical image. Simultaneously, the frequency of generating target images needs to be denser. As training progresses and the deformation field becomes more stable, the noise intensity and frequency of generating target images decrease accordingly. This hierarchical scheduling approach ensures that the final canonical image approaches the quality of per-step updates while speeding up the training process by 14 times (4.8 hours to 20 minutes.) We opt for diffusion loss over SDS because using SDS, as mentioned in the Reconfusion [71] paper, is more prone to generate artifacts.

### 3.3   Separated NaRCan

When encountering overly complex scenes, relying solely on a single natural canonical image representing the entire scenario is impractical and unrealistic. Hence, we need to segment the original video into multiple segments $\{S_1, ..., S_k\}$ and train dedicated residual deformation MLPs $\{R_1, ..., R_k\}$ for each segment to obtain k natural canonical images $\{C_1, ..., C_k\}$. It is worth noting that $S_i$ and $S_{i+1}$ have an overlap, referred to as the overlap window. Frames within this region are obtained using linear interpolation shown in Figure 4. This method ensures that excellent temporal consistency is maintained when switching from canonical image $C_i$ to $C_{i+1}$. Table 1 demonstrates that our temporal consistency surpasses all existing methods even after segmentation. Additionally,

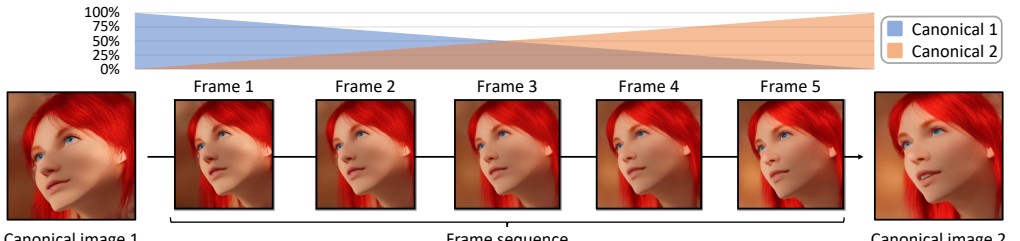

Figure 4: **Linear interpolation.** After using the grid trick [18] to obtain the highly consistent canonical images $C_k$ and $C_{k+1}$, we interpolate all frames within the overlap window. As time progresses, the weight for reconstructing each frame gradually shifts from referencing $C_k$ to solely referencing $C_{k+1}$. We achieve editing results with remarkable temporal consistency through this linear interpolation approach. Please refer to our supplementary material for more video results.

we adopt different processing approaches for various downstream tasks to ensure that Separated NaRCan can adeptly adapt to these tasks.

**Style transfer.** With multiple canonical images obtained, we utilize the grid trick [14, 24] to ensure sufficient consistency in style and content across the $k$ canonical images. Specifically, we concat the canonical images into a larger image (with $2 \times 2$ canonical images) and use ControlNet [75] to perform text-guided image editing.

**Video editing.** When addressing video editing tasks such as handwriting, we leverage the pre-trained optical flow models to compute the flow between the k canonical images and use this flow to warp the editing content from $C_1$ to the $C_k$ image.

## 4 Experiments

### 4.1 Experimental Setup

We conduct experiments to underscore the robustness and versatility of our proposed method. Our representation is robust with a variety of deformations, encompassing rigid and non-rigid objects, as well as complex scenarios such as smog and waves. We commence the introduction of the diffusion model at the 1000th iteration. From iteration 1000 to iteration 3000, the noise intensity is set at 0.4, and the target image generation frequency is every 10 iterations. Subsequently, spanning from iteration 3001 to iteration 5000, the noise intensity is adjusted to 0.3, with the target image generation frequency occurring every 100 iterations. Beyond the 5000th iteration mark, the noise intensity decreases to 0.2, and the target image is generated every 2000 iterations. The total iteration is 12000 iterations, and Figure 3 is shown to visualize the process of noise scheduling. On a single NVIDIA RTX4090 GPU, the average training duration is approximately 20 minutes when utilizing 100 video frames. When evaluating the temporal consistency in Table 1, we compare our separated NaRCan with other compared methods by setting $k = 3$, *i.e.*, we represent the sequence using three canonical images. By adjusting the training parameters accordingly, the optimization duration can be varied from 20 minutes to an hour.

### 4.2 Evaluation

**Video editing.** We run our method CoDeF [45], Hashing-nvd [5], CCEdit [15], and MeDM [13] on the DAVIS [49] and BalanceCC Benchmark that CCEdit proposed for evaluating the results of video editing. In Figure 5, we show the comparison of visual results. The BalanceCC Benchmark provides a unified text prompt for each scene, and we ensure that all video clips from this benchmark are restricted to a maximum of 100 frames. We utilize ControlNet [75] to edit the video using the unified text prompt provided by the BalanceCC Benchmark for a fair comparison. We executed MeDM and CCEdit within their provided environment settings and pre-trained models. Moreover,

Table 1: **Quantitative results on the BalanceCC [15] dataset.** There are 100 videos in BalanceCC. To ensure a representative distribution similar to BalanceCC, we randomly select 50 videos from BalanceCC and calculate warping and interpolation errors, which are the metrics for temporal consistency. Our method outperforms these baseline methods in terms of temporal consistency.

| Method | Venue | Short-term $E_{warp}\downarrow$ | Long-term $E_{warp}\downarrow$ | Interpolation error $\downarrow$ |
|---|---|---|---|---|
| Hashing-nvd [5] | ICCV 2023 | 0.0070 | 0.0495 | 9.204 |
| CoDeF [45] | CVPR 2024 | 0.0085 | 0.0785 | 8.721 |
| MeDM [13] | AAAI 2024 | 0.0072 | 0.0583 | 9.941 |
| Ours | - | **0.0065** | **0.0484** | **8.365** |

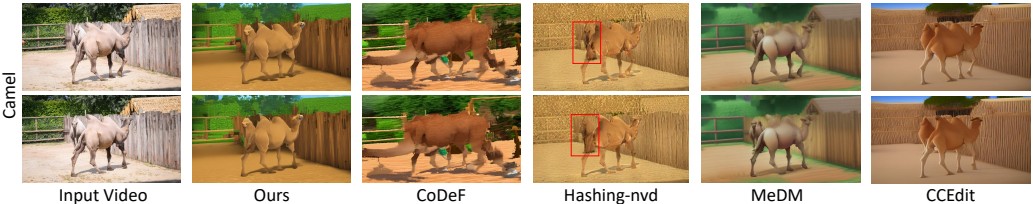

Text Prompt: A camel walking in an enclosure with a wooden fence and greenery in the background, Minecraft world style.

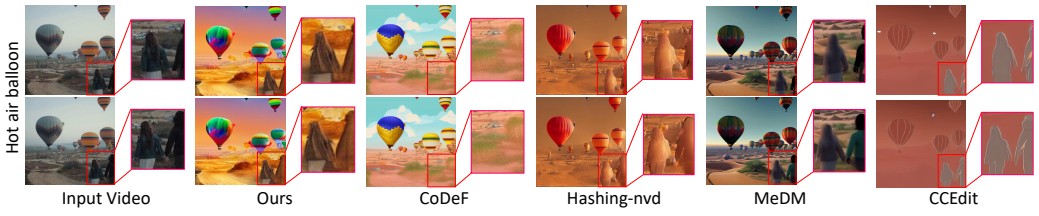

Text Prompt: Hot air balloons adrift over an ancient desert, Chibi Animation style.

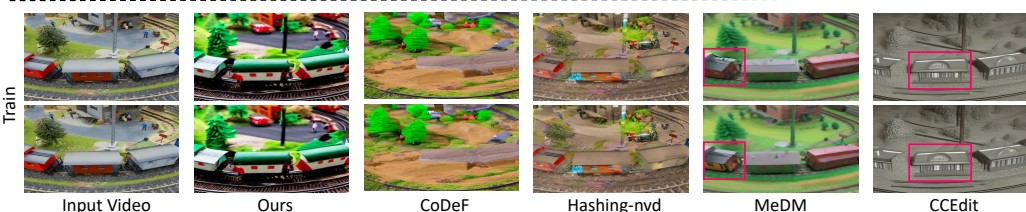

Text Prompt: A model train moving along the track with miniature figures and trees alongside, all depicted in a claymation style.

Figure 5: **Qualitative comparisons on text-guided video-to-video translation.** Our method achieves prompt alignment, synthesis quality, and temporal consistency best. Zoom in for the best view, and please refer to the supplementary materials for video comparisons. (a) In the camel scene, Medm [13] fails to generate clear-textured images to ensure temporal consistency, while CCEdit [15] fails to correctly identify the second camel in the background. (b) CoDeF [45] misses capturing the presence of a person in the bottom right corner, Hashing-nvd [5] exhibit noticeable contours due to masking, and both MeDM and CCEdit suffer from temporal inconsistency issues. For instance, in MeDM, the person transitions from wearing black clothes to blue clothes. (c) MeDM and CCEdit still exhibit temporal inconsistency issues, such as significant color, texture, and structure changes. Other methods almost entirely lose the original train information or appear as unnatural artifacts.

Hashing-nvd is a video decomposition work that outputs two images representing foreground and background. To maintain the consistent style of these two images for video editing, We utilize the grid trick proposed by RAVE [24] to tackle this problem. Finally, there are also some video-editing results on DAVIS [49], and the corresponding text prompts originated from the BalanceCC Benchmark. For the evaluation of temporal consistency, as shown in Table 1, we utilize short-term [29, 77] and long-term [30] warping errors, along with interpolation error [32], as our primary metrics.

**Metrics for evaluation.** Since our main focus is text-guided video editing, we conduct user studies on edited videos compared with other methods, like CoDeF [45], MeDM [13], and Hashing-nvd [5].

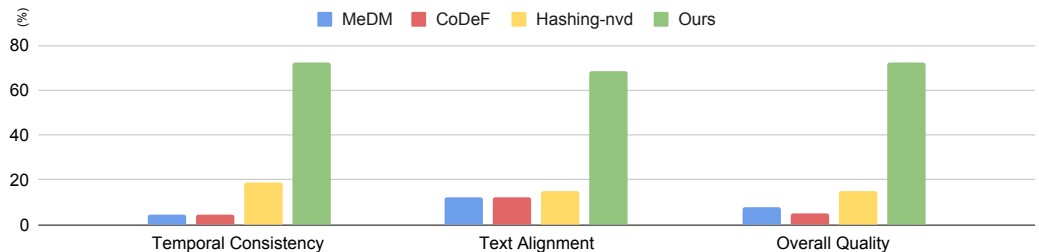

Figure 6: **User Study.** Our method achieves the highest user preference ratios across all three aspects, compared with MeDM [13], CoDeF [45], Hashing-nvd [5].

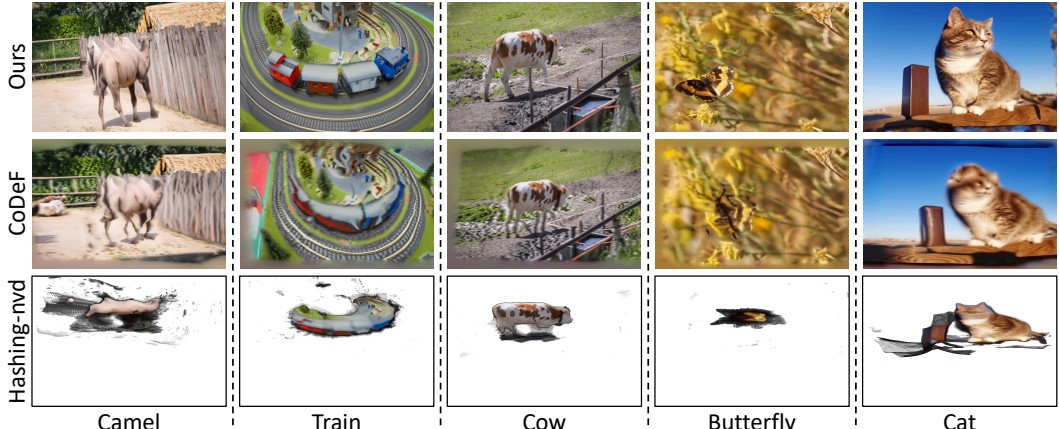

Figure 7: **Qualitative comparisons on the canonical image.** Our method generates more natural canonical images through a fine-tuned diffusion prior compared with CoDeF [45], Hashing-nvd [5]. The capability of the canonical image to represent input frames plays a crucial role in downstream applications. (Hashing-nvd consists of two canonical images. Here, we have selected the canonical image representing the foreground.)

In the user study, 36 participants were shown two edited videos, the original video on each page, and the text prompt used to edit the videos. There are three critical questions for users to answer. (1) *Which video has better temporal consistency?* (2) *Which video aligns better with the text prompt?* (3) *Which video has better overall quality?* Figure 6 summarizes the user study results, demonstrating that our method outperforms in all three dimensions.

**Comparison of canonical images.** In Figure 7, we run our method, CoDeF [45], and Hashing-nvd [5] on DAVIS [49] and BalanceCC [15]. Note that these are the canonical images of the input video, and we show the foreground atlas for Hashing-nvd. There will be two output atlases for the background and foreground of Hashing-nvd. For comparison, we only show the foreground atlas for Hashing-nvd. As a result, shown in Figure 7, the output canonical images of our method are more natural than others, even in the scenes with the dramatic motion of the objects, such as scenes named "train" and "butterfly." We can clearly see that our canonical images preserve the original object information well in the above two scenarios.

**Downstream video processing.** To evaluate the performance of our method's handwritten video editing, we compare with CoDeF [45] and Hashing-nvd [5], which produce canonical images of the video for users to write the characters on the canonical image to accomplish video editing. We write "NaRCan" to test the performance of these three methods on two scenes, "gold-fish" and "train", in the BalanceCC Benchmark [15] in Figure 8(a). We extract the same frame of videos for comparison.

For dynamic video segmentation, we segment the mask using the Segment Anything Model (SAM) [28] based on the learned canonical image of each method and propagate it to the sequence.

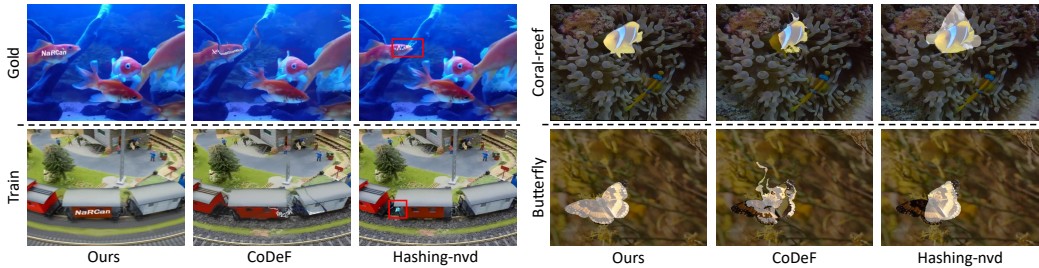

(a) Adding handwritten characters       (b) Dynamic video segmentation.

Figure 8: **Qualitative comparisons on (a) adding handwritten characters and (b) dynamic video segmentation.** Our method represents a natural image via diffusion prior, thus can achieve temporally consistent video editing and able to precisely edit desired areas.

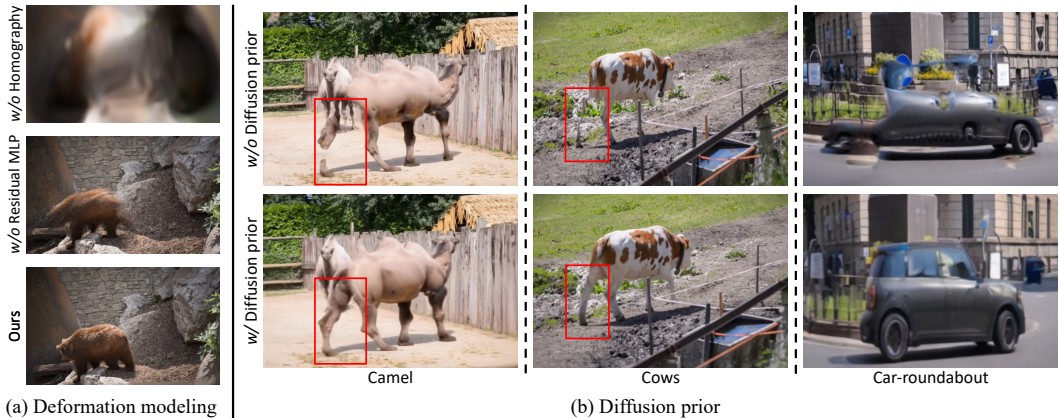

(a) Deformation modeling       (b) Diffusion prior

Figure 9: **Ablation studies.** (a) Deformation modeling: (*Top*) We show that canonical images without homography modeling fail to generate a faithful image as the capacity of residual deformation MLP could dominate the training process and still achieve near-perfect frame reconstruction. (*Mid*) On the contrary, without residual deformation MLP, our method cannot model local non-rigid transformation, resulting in blurry foreground objects. (*Bottom*) Combining homography and residual deformation MLP has the best of both worlds and achieves the best canonical image representation. (b) Diffusion prior: (*Top*) Without diffusion prior to regularizing the canonical image, the training process relies only on the frame reconstruction and could sacrifice the faithfulness of the canonical image. (*Bottom*) Our fine-tuned diffusion prior effectively corrects the canonical image to faithfully represent the input frames and results in natural canonical images.

The target of the mask in these two scenes are the clownfish named "coral-reef" and the flying butterfly in the scene called "butterfly." We use white to mark the mask for better visibility in Figure 8(b).

## 4.3 Ablation Study

**Homography & Residual Deformation MLP.** In this section, we conducted ablation experiments focusing on both homography and residual deformation MLP. Figure 9(a) clearly illustrates that using only MLP to fit the deformation field [31, 46, 47, 51] results in unsuitable canonical images for downstream tasks. Without the global information homography provides, the model encounters difficulties in converging the diffusion loss and instead focuses solely on optimizing the reconstruction loss. Conversely, if we rely solely on homography to express the deformation field, homography's expressive power is limited in capturing detailed variations in non-rigid objects. As a result, only approximate and blurred outcomes can be obtained.

**Diffusion prior.** Relying on homography and the residual deformation MLP achieves relatively better canonical images than previous methods. However, the lack of assistance from the diffusion prior still prevents the stable generation of high-quality natural canonical images. Figure 9(b) demonstrates the significance of supervising canonical images with the diffusion prior.

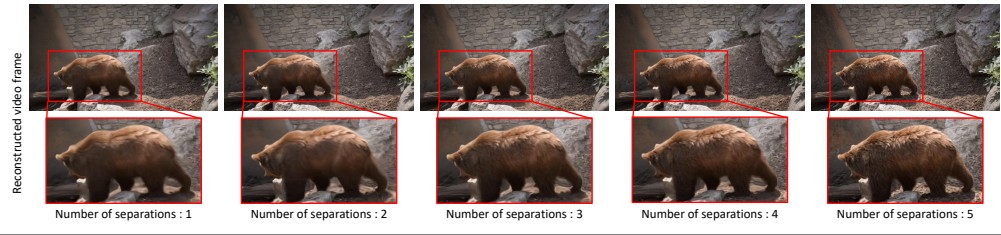

Reconstructed video frame

| Number of separations : 1 | Number of separations : 2 | Number of separations : 3 | Number of separations : 4 | Number of separations : 5 |

| N of separations | Training time(s) | PSNR ↑ | SSIM ↑ | short-term $E_{warp}$ ↓ | long-term $E_{warp}$ ↓ |
|:---:|:---:|:---:|:---:|:---:|:---:|
| 1 | 771.50 | 23.814 | 0.6369 | **0.0016** | **0.0304** |
| 2 | 1530.79 | 24.423 | 0.6762 | 0.0019 | 0.0310 |
| 3 | 2275.20 | 24.852 | 0.7017 | 0.0021 | 0.0312 |
| 4 | 3015.32 | 25.185 | 0.7191 | 0.0022 | 0.0326 |
| 5 | 3761.40 | **25.398** | **0.7301** | 0.0022 | 0.0334 |

Figure 10: **Trade-off between reconstruction quality and temporal consistency with varying separations.** The visual results demonstrate that increasing the number of separations (from 1 to 5) improves the reconstruction quality of video frames. However, the table reveals a trade-off: as the number of separations increases, temporal consistency decreases. Our empirical findings suggest that using 3 separations typically achieves a balance between reconstruction quality and temporal consistency for most scenarios.

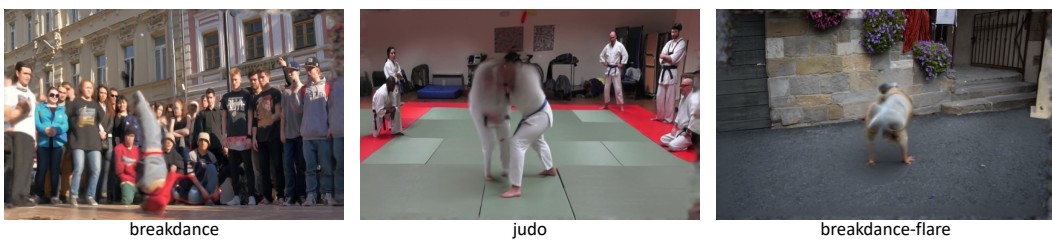

| breakdance | judo | breakdance-flare |

Figure 11: **Failure cases.**

**Impact of Separation Numbers.** To investigate the optimal configuration of our Separated NaRCan, we conduct ablation studies by varying the number of canonical images from one to five. Our analysis reveals a clear trade-off between reconstruction quality and temporal consistency, as illustrated in Figure 10. While increasing the number of separations improves frame reconstruction fidelity by allowing more detailed scene representation, it introduces additional transition regions that may compromise temporal coherence. Through extensive experimentation, we find that using three canonical images strikes an optimal balance between representational capacity and temporal stability for most video scenarios.

## 5 Conclusion

In this paper, we introduce NaRCan, a video editing framework, integrating diffusion priors and LoRA [18] fine-tuning to produce high-quality natural canonical images. This method tackles the challenges of maintaining the natural appearance of the canonical image and reduces training times with new noise-scheduling techniques. The results show NaRCan's advantage in managing complex video dynamics and its potential for wide use in various multimedia applications.

**Limitations.** Our method relies on LoRA [18] fine-tuning to enhance the diffusion model's ability to represent the current scene. However, LoRA fine-tuning is time-consuming (about 30 minutes). Additionally, our training pipeline includes diffusion loss, which increases the training time. In cases of extreme changes in video scenes, diffusion loss sometimes fails to guide the model in generating high-quality natural images (Figure 11). These limitations point out the challenge of balancing model adaptability with computational efficiency and effectiveness in varied conditions.

## Acknowledgments and Disclosure of Funding

This research was funded by the National Science and Technology Council, Taiwan, under Grants NSTC 112-2222-E-A49-004-MY2 and 113-2628-E-A49-023-. The authors are grateful to Google, NVIDIA, and MediaTek Inc. for generous donations. Yu-Lun Liu acknowledges the Yushan Young Fellow Program by the MOE in Taiwan.

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

# A Appendix

## A.1 Single NaRCan compared with Separated NaRCan

**Dissecting canonical-based methods.** In this section, we will delve deeper into two different canonical-based methods: CoDeF and Hashing-nvd. As shown clearly in Figure 12, CoDeF generates poor-quality canonical images due to the lack of supervision and constraints from the diffusion prior. In contrast, our method can consistently produce high-quality canonical images regardless of the video length, effectively preparing for subsequent downstream editing tasks. The drawbacks of Hashing-nvd are also apparent. This method relies heavily on the Mask-RCNN technique, often resulting in inaccurate or incorrect foreground and background segmentation. Consequently, the final canonical images generated are difficult to edit or inapplicable to techniques such as ControlNet.

**Parameter settings of Separated NaRCan.** Subsequently, as shown in Figure 13, From our experiments, we found that when using Separated NaRCan, it is crucial to limit the number of segmentations in the video. This is because the editing information on the edited canonical image is propagated through warping using flows. If the flow is inaccurate or contains errors, excessive warping can lead to severe cumulative errors, significantly damaging the edited content.

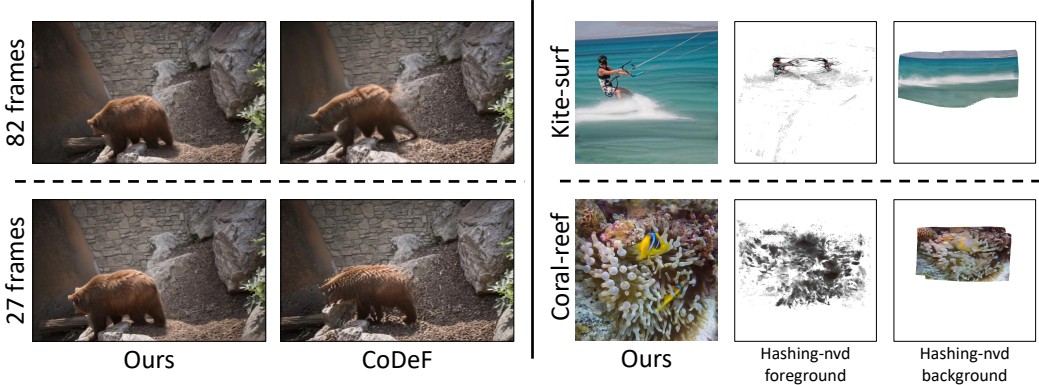

Figure 12: **Canonical analysis.** (a) Compared to existing canonical-based methods, our approach robustly generates high-quality natural canonical images regardless of the video length. (b) Our method accurately preserves the correct foreground and background information from the original scenes, avoiding severe distortions or warping and preventing generating content inconsistent with the scene. For example, in "kite-surfing", Hashing-nvd erroneously generates an additional person.

## A.2 Video Comparisons

We provide an interactive HTML interface to browse video results for comparisons. Specifically, we provide video comparisons on three different tasks: (1) ControlNet style transfer, (2) dynamic segmentation, and (3) adding handwritten characters. We compare our proposed method, NaRCan, with state-of-the-art methods: Hashing-nvd [5], CoDeF [45], and MeDM [13]. We also visualize the optimized canonical images if available for reference.

## A.3 User Study

To conduct our user study, we employ GitHub Pages in conjunction with a Google Form to facilitate user evaluations of video quality. Each evaluation session comprises 15 scenes, each of which contains 3 questions. For these evaluations, we randomly select 15 scenes from a pool of 100 scenes in BalanceCC Benchmark [15]. Each evaluation page (Figure 14) presents a video edited using our method alongside a randomly selected video sourced from compared methods: MeDM [13], CoDeF [45], or Hashing-nvd [5].

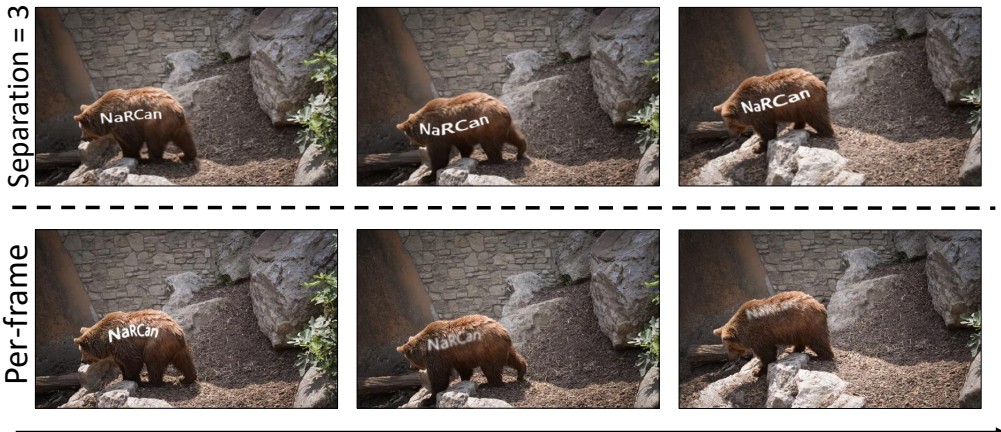

Figure 13: **Separated NaRCan and per-frame warping.** Since we have the flexibility to separate into multiple canonical images, we conduct an experiment to determine how many canonical images are optimal. We test using Separated NaRCan with segmentation equal to 3 and segmentation equal to N, where N equals the number of frames in the video. The results show that the editing information is damaged and has significant displacement due to inaccurate optical flow.

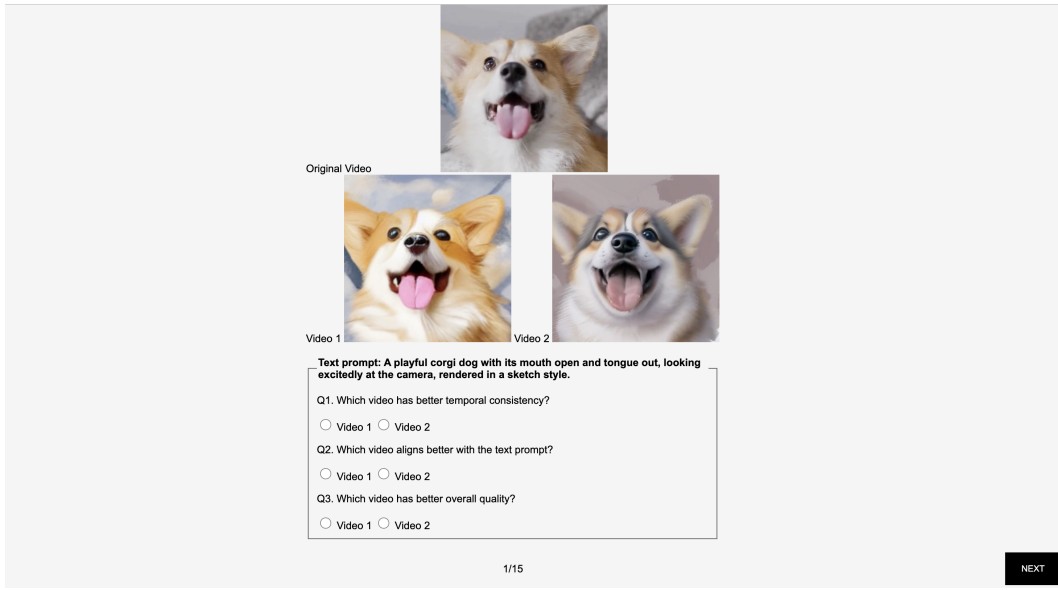

Figure 14: **User Study Website.** The video above is the original video. The two videos below are the ones being compared. We ask participants to determine which of the given videos best matches the description provided in the questions.

## A.4   Grid Trick

We adopt the technique named "grid trick" from RAVE [24]. In that paper, a method called grid trick is introduced, where multiple images are merged together and then fed to ControlNet [75] for editing to obtain style transfer images with sufficient consistency in content and color tones. Therefore, when using Separated NaRCan, we only need to apply the grid trick to our k canonical images to achieve this desired consistency among the canonical images easily [24].

Finally, coupled with the linear interpolation method mentioned in our paper, we can ensure that the video maintains sufficient temporal consistency, thereby successfully creating a high-quality style transfer video.

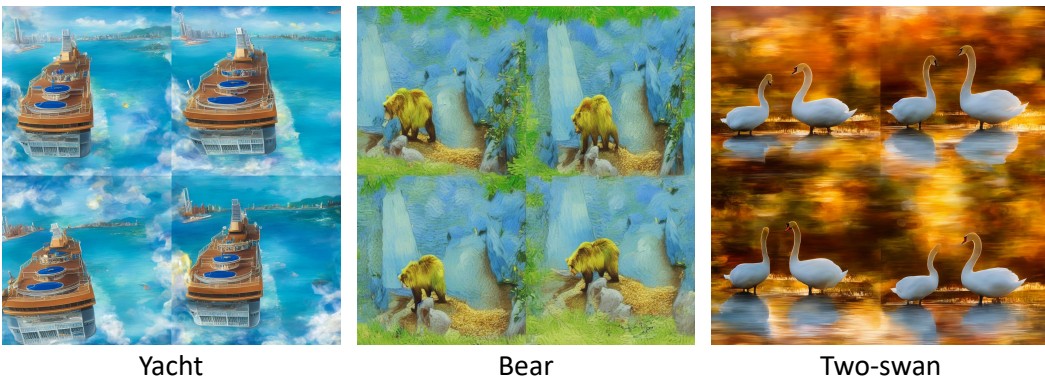

| Yacht | Bear | Two-swan |

Figure 15: **Canonical images after using the grid trick.** Using Separated NaRCan, we will obtain multiple canonical images, and through the grid trick, we can generate a high-quality and consistent style transfer canonical image. Therefore, Separated NaRCan still demonstrates excellent performance in this task.

