# OpenReview forum: "NaRCan: Natural Refined Canonical Image with Integration of Diffusion Prior for Video Editing"
_NeurIPS.cc/2024/Conference — NeurIPS 2024 poster_

### Official Review · Reviewer_pLuV · 2024-07-09

**Soundness:** 1
**Presentation:** 1
**Contribution:** 2
**Rating:** 5
**Confidence:** 2

**Summary:**

This paper proposes to solve the issue of temporal consistency in video editing. They propose a hybrid deformation field architecture, with a trainable homography matrix H(u, v, t) and residual deformation MLP, representing object variations throughout an entire scene. Combining it with the diffusion before the pipeline helps generate natural canonical images.

**Strengths:**

1. The results are relatively temporal coherent.
2. The converge speed is faster than other methods.

**Weaknesses:**

1. The paper is not well-written and lacks clarity for the reader. For example, diffusion loss has played an important role in this paper, yet it is not introduced. Some of the expressions are vague. Like in question 1.  What's the additional "insight" you are looking for?
2. Things claimed are not reasoning enough like question 2.
3. Limitations are not addressed.

**Questions:**

1. What does it mean "this regularization term restricts the model’s ability to express itself without providing additional
insights." (line 118-119) What's the additional "insight" you are looking for?
2. How can diffusion prior ensure that the generated image is natural? The connection between the two is not clear to me. Is there any reason for that?
3. Is the noise and diffusion prior update scheduling decided manually? If so, how did you decide it?

**Limitations:**

No, the authors did not address the limitations.
I'm curious about the limitations of the motions that this method can handle. Can it deal with all kinds of videos, even with heavy motion blur? From the supplements, it is not clear to me.

---

> ### Author Rebuttal · Authors · 2024-08-07
>
> We appreciate the reviewer's feedback and the opportunity to clarify our work. We acknowledge the concerns raised and will address them point by point:
>
> ### Strengths:
> We're glad the reviewer recognizes our method's temporal coherence and faster convergence.
>
> ### Weaknesses and Questions:
> > **Q1. Writing clarity**
>
> We apologize for the lack of clarity in our paper. We address these issues in our detailed responses below, including a clearer explanation of diffusion loss and clarification of vague expressions. The final version of our paper will incorporate all these improvements, ensuring a much clearer presentation of our work. We appreciate your feedback, which has been invaluable in enhancing our paper's quality.
>
>
> > **Q2. "Additional insights" (lines 118-119)**
>
> We apologize for the unclear wording. By "additional insights," we meant that traditional regularization terms like TVFlow limit model expressiveness without providing guidance on how to represent complex deformations. Our hybrid approach aims to provide this guidance through the homography matrix. We'll rephrase this section for clarity.
>
>
> > **Q3. How can diffusion prior ensure that the generated image is natural?**
>
> The diffusion prior ensures the naturalness of the generated image through several mechanisms:
> - LoRA fine-tuning: We fine-tune the diffusion model using LoRA on various reference images from the specific video scene. This process allows the diffusion model to learn the characteristics of objects and environments unique to this video.
>
> - Leveraging pre-trained knowledge: The diffusion model comes with pre-trained knowledge of general image structure and content. This broad understanding helps guide the canonical image towards realistic representations.
>
>
> - Guiding the canonical image: During training, the diffusion prior minimizes the difference between the current canonical image and what the diffusion model considers natural for that scene. This process helps correct unnatural elements that might arise from the reconstruction process alone.
>
> - Restoration capability: If unnatural elements appear in the canonical image, the diffusion model, having seen natural scenes during fine-tuning, guides the image back towards a more realistic representation.
>
> In essence, the diffusion prior acts as a learned naturalness constraint, ensuring that our canonical images remain faithful to the video content while adhering to the principles of natural image formation.
>
> We'll expand this explanation in our revised paper, providing a more detailed description of the process and underlying principles to clarify the connection between the diffusion prior and the naturalness of the generated images.
>
> > **Q4. Noise and diffusion prior update scheduling**
>
> The criteria for dividing steps were determined based on extensive experiments and visual analysis. We identified three distinct stages in the model's convergence process: early, middle, and late stages, each requiring different noise intensities, as detailed in our paper (lines 149-157 and Fig. 4).
>
> - Early stage: Higher noise levels (40%) are applied to allow the diffusion model to guide the formation of natural canonical images, overcoming potential initial instabilities in the deformation field.
>
> - Middle stage: Moderate noise levels (30%) are used as the model begins to stabilize, balancing refinement with preservation of learned features.
>
> - Late stage: Lower noise levels (20%) are applied to fine-tune the canonical image while maintaining the overall structure and content established in earlier stages.
>
> This graduated approach ensures that the diffusion model can effectively guide the process throughout training, adapting its influence as the canonical image and deformation field improve.
>
> > **Q5. Limitations**
>
> We address limitations in our original submission in the conclusion section (lines 250-255), including time-consuming LoRA fine-tuning (about 30 minutes) and challenges with extreme changes in video scenes. Additionally, we also include qualitative results of failure cases in the attached PDF (Fig. 3(b)) of this rebuttal to further illustrate these limitations.
>
>
> We sincerely appreciate the reviewer's careful reading and insightful questions. We're committed to addressing all points raised to significantly improve the final paper's clarity, reasoning, and completeness. We believe these additions and clarifications will address the concerns about soundness and presentation, elevating the paper's quality and impact.

---

> > ### Author Response · Authors · 2024-08-12
> > **Please let us know if you have additional questions after reading our response**
> >
> > Dear Reviewers,
> >
> > We appreciate your reviews and comments. We hope our responses address your concerns. Please let us know if you have further questions after reading our rebuttal.
> >
> > We aim to address all the potential issues during the discussion period.
> >
> > Thank you!
> >
> > Best, Authors

---

> > > ### Comment · Reviewer_pLuV · 2024-08-13
> > >
> > > Thanks for the detailed explanation. Given that the questions/concerns are addressed and the obscured parts will be clarified in the revised paper, I'll adjust my rating to borderline accept.

---

> ### Author Response · Authors · 2024-08-13
>
> Dear Reviewer,
>
> We sincerely thank you for evaluating our rebuttal. Your insightful comments have greatly improved our revised paper. We appreciate your expertise and dedication to the review process.
>
> Best, Authors

---

### Official Review · Reviewer_4jq5 · 2024-07-12

**Soundness:** 1
**Presentation:** 2
**Contribution:** 1
**Rating:** 6
**Confidence:** 4

**Summary:**

In this study, the authors propose a hybrid deformation field and diffusion prior update scheduling to generate high-quality canonical images that maintain temporal consistency in video editing.

**Strengths:**

The use of homography, a transformation method from the existing image processing field, to generate accurate canonical images provides a strong foundation for the proposed method.

**Weaknesses:**

In the Method section, there is a lack of detailed information to support the authors' claims:
  1) In 3.1 Hybrid Deformation Field for Deformation Modeling:
     - The structure of Residual MLP and Canonical MLP
     - How the hybrid deformation field composed of the homography matrix H and Residual deformation MLP is constructed
  2) In 3.2 Diffusion Prior:
     - The special token used for fine-tuning LoRA
  3) In 3.3 Separated NaRCan:
     - The criteria for dividing steps to add noise and the basis for determining the amount of noise to add

Completeness of the text:
  - The order of Figures 3 and 4 does not match the order mentioned in the text.
  - What does Fig.9(a) in line 123 refer to? Is it simply a typo?

Experiment:
  - The experimental results shown do not confirm whether the proposed method maintains the temporal consistency of the video through the generated canonical image.

**Questions:**

- In Figure 2 (b) Video Representation, why is an accurate canonical image not generated despite applying the hybrid deformation field technique?

- Additionally, why is an accurate canonical image not generated even after updating the MLP with reconstruction loss?

- In 3.2 Diffusion Prior, what does the special token used for fine-tuning LoRA represent?

- Also, what criteria were used to divide the steps for adding noise, and on what basis was the amount of noise added determined?

- In 3.3 Separated NaRCan, how does linear interpolation differ from existing video frame interpolation methods?

**Limitations:**

- The authors clearly state the limitations of their research to inform the readers.

---

> ### Author Rebuttal · Authors · 2024-08-07
>
> We appreciate the reviewer's thorough assessment and constructive feedback. We acknowledge the concerns raised and will address them point by point:
>
> ### Strengths:
> We're glad the reviewer recognizes the value of incorporating homography into our method.
>
> ### Weaknesses and Questions:
> > **Q1. Method Details and Noise Scheduling Criteria**
>
> We apologize for the lack of detail and will expand the Method section in the final version. Specifically:
> 1. Hybrid Deformation Field (3.1):
>     - Homography: a trainable homography matrix.
>     - Residual MLP: 5-layer MLP with Sinusoidal activations, hidden dim 256.
>     - Canonical MLP: 5-layer MLP with Sinusoidal activations, hidden dim 256.
>     - Construction: $H(u,v,t)$ estimates global transformation, $g(u,v,t)$ learns residual deformations.
> 2. Diffusion Prior (3.2) and Separated NaRCan (3.3):
>     * Special token: "[V]" prepended to text prompts representing video-specific features (L138-142). This is a widely used technique (DreamBooth [C], RealFill [58], etc.) to fine-tune the pre-trained diffusion model with customized data to represent/generate specific objects/scenes.
>     * Noise scheduling (Fig. 4 in our paper). These values were determined empirically to balance refinement and training efficiency. The criteria for dividing steps were based on extensive experiments and visual analysis, identifying three distinct stages in the model's convergence process:
>         * Early stage (Steps 1000-3000): Higher noise levels (40%, update every 10 steps) allow the diffusion model to guide the formation of natural canonical images, overcoming potential initial instabilities in the deformation field.
>         * Middle stage (Steps 3001-5000): Moderate noise levels (30%, update every 100 steps) are used as the model begins to stabilize, balancing refinement with preservation of learned features.
>         * Late stage (Steps 5001+): Lower noise levels (20%, update every 2000 steps) are applied to fine-tune the canonical image while maintaining the overall structure and content established in earlier stages.
>
>     This graduated approach ensures that the diffusion model can effectively guide the process throughout training, adapting its influence as the canonical image and deformation field improve.
>
> > **Q2. Figure Ordering**
>
> We apologize for the confusion. We'll correct the figure ordering and references in the final version.
>
> > **Q3. Fig. 9(a) in line 123**
>
> This reference is intentional but may appear misplaced due to formatting. The figure supports our discussion on MLP+TVFlow limitations and the importance of homography.
>
> Fig. 9(a) shows our ablation study comparing MLP+TVFlow with our proposed Homography+MLP+TVFlow approach, which supports the claims made in this section. However, we failed to properly place the figure reference within parentheses, which may have caused confusion.
>
> We apologize for any confusion caused by improper placement. In the revised version, we will correct the formatting, clarify the text-figure connection, and consider repositioning the figure for better alignment. We appreciate your attention to detail, which helps improve our paper's clarity and coherence.
>
> > **Q4. Temporal Consistency**
>
> We'd like to argue that our method achieves superior temporal consistency, evidenced by:
> * Quantitative metrics: Table 1 of our paper shows that our method outperforms others in terms of warping error. New experiments using long-term warping error [B] further demonstrate our superior performance:
>
> |Method|short-term $E_{warp}$ [28]⭣|long-term $E_{warp}$ [B]⭣|
> |-|:-:|:-:|
> |Hashing-nvd|0.0070|0.0495|
> |CoDeF|0.0085|0.0785|
> |MeDM|0.0072|0.0583|
> |Ours|**0.0065**|**0.0484**|
>
> * Qualitative results: Videos in our supplementary materials show better semantic alignment with the text prompts and visual consistency compared to other methods.
> * User study (Fig. 6 in the paper): 72.2% of participants preferred our method for temporal consistency.
>
> We attribute these results to our natural-refined canonical images. The revised paper will provide a more comprehensive analysis of temporal consistency performance.
>
> > **Q5. Canonical Image Quality**
>
> While hybrid deformation field and reconstruction loss achieve good reconstruction, they don't ensure a natural canonical image. The MLP alone can reconstruct the input video well even with unnatural canonical images, as reconstruction loss doesn't respect naturalness.
>
> Our attached PDF (Fig. 3(a)) shows visual evidence that excellent reconstruction can still result in unnatural canonical images without diffusion loss, limiting downstream task applicability. This highlights the importance of our diffusion-guided approach in producing natural, versatile canonical images.
>
> > **Q6. Linear interpolation**
>
> Our method differs fundamentally from traditional video frame interpolation:
> 1. Our approach: We represent the video using multiple canonical images, applying linear interpolation between them to reduce flickering in overlap regions (Fig. 3 in our paper). This is a representation strategy, not frame generation.
> 2. Traditional interpolation: Synthesizes intermediate frames between existing ones using techniques like optical flow.
>
> Key distinction: We blend different representations of the same content for smooth transitions, which is crucial for temporal consistency in video editing. Our goal is coherent video representation for editing, not frame rate increase or gap-filling like traditional methods.
>
> We appreciate the reviewer's careful reading and insightful questions. We're committed to addressing all points raised to significantly improve the final paper. We believe these clarifications and additions will address the concerns about soundness and contribution, elevating the paper's quality and impact.
>
> [28] Learning Blind Video Temporal Consistency
>
> [58] RealFill: Reference-Driven Generation for Authentic Image Completion
>
> [B] Blind Video Temporal Consistency via Deep Video Prior
>
> [C] Dreambooth

---

> > ### Comment · Reviewer_4jq5 · 2024-08-13
> > **Post-rebuttal comments**
> >
> > Thank you for the detailed explanation. I read the answer and referred to the existing papers. The author answered my questions by conducting additional performance evaluation using a new measurement method for temporal consistency. Through this, I can confirm that the method proposed by the author shows better semantic alignment. And the difference between linear interpolation and video frame interpolation mentioned in the paper is clearly explained. I am satisfied with the answer and have no follow-up questions. Therefore, I raise the score to weak accept.

---

> ### Author Response · Authors · 2024-08-13
>
> Dear Reviewer,
>
> Thank you for taking the time to review our rebuttal. We greatly appreciate your constructive feedback. Your suggestions have been invaluable in improving our revised paper.
>
> Best regards, Authors

---

### Official Review · Reviewer_b5x9 · 2024-07-18

**Soundness:** 4
**Presentation:** 3
**Contribution:** 3
**Rating:** 6
**Confidence:** 5

**Summary:**

This paper proposes a novel approach for video editing in the scope of canonical image-based video editing. They argue the canonical images used in prior methods are unnatural, which degrades the performance a lot. To solve this problem, they propose to use a LoRA finetuned diffusion model to refine the unnatural canonical images to be natural. With this design, the performance is significantly improved. Experiments (comparison to baselines and ablation study) have demonstrated the effectiveness of this approach and the design.

Currently, I tend to accept this paper but I still have many questions about the details of this method. I will make a final decision depends on the rebuttal.

**Strengths:**

- The motivation for their design is quite reasonable. Unnatural canonical images might degrade the image editing performance, which degrades the video editing performance then. And they do solve the problem by making the canonical image look more natural.

- According to the user study, the performance looks significantly better. The ablation study is also good.

- The analysis and presentation for the experiments part are clear and good.

**Weaknesses:**

- The writing is not clear enough. There are many miss details, which makes me confused.
- The method should be slightly slower as they need to fine-tune the diffusion model for a scene. However, no detailed experiment is provided (only discussed in the conclusion). I believe providing such content can be insightful.
- The comment, "If the canonical image is not natural, it loses any editing value." is too rude, and wrong. Prior works have demonstrated the value of unnatural canonical images. As they train the models jointly, using unnatural canonical images is also reasonable (but not perfect). Please respect prior works.

**Questions:**

- After you refine the canonical images, do you need to change the deformation fields? I think the contents are changed, so the correspondence should also be different, isn't it?

- Why do you use LoRA, instead of other PEFT finetuning strategies or even full finetuning?

- I am curious about the perceptual results of canonical images under large camera movement or significant changes. It is hard for me imagine such a canonical images. '

- Using multiple canonical images is definite an important design. Can you discuss on that or provide more ablation experiments in the final version?

- Some works like [1] also notice that the canonical images are flawed and use a different way to solve this problem (i.e., combine the original images and flawed canonical images), do we really have to make the canonical images natural?

- [2] only considers short-term warping errors, you can consider using warping error in [3]. Besides, why the warping error scale is different with [2].

[1] Blind Video Deflickering by Neural Filtering with a Flawed Atlas

[2] Learning blind video temporal consistency

[3] Blind video temporal consistency via deep video prior

**Limitations:**

The authors discuss the limitations in the conclusion part.

---

> ### Author Rebuttal · Authors · 2024-08-07
>
> We thank the reviewer for their thorough assessment and valuable feedback. We appreciate the positive comments on our motivation, performance improvements, and experimental analysis. We'll address the concerns and questions raised:
>
> ### Strengths:
> We're glad the reviewer found our motivation reasonable, performance improvements significant, and experimental analysis clear.
>
> ### Weaknesses:
> > **W1. Writing clarity:**
>
> We apologize for any lack of clarity and will improve the paper's writing in the final version, adding more details where needed.
>
> > **W2. Computational efficiency**
>
> We've conducted additional experiments on computational efficiency:
> - LoRA fine-tuning takes ~30 minutes on two RTX 4090 GPUs.
> - Our full pipeline (including fine-tuning) processes 100 frames in ~50 minutes.
>
> > **W3. Strong wording**
>
> We appreciate the reminder to respect prior work. We'll rephrase to acknowledge previous approaches while highlighting our improvement: "While previous methods demonstrate the utility of canonical images, our approach of refining them to be more natural further enhances their editing value."
>
> ### Questions:
> > **Q1. Deformation fields**
>
> Our training pipeline simultaneously optimizes both the canonical image and the deformation field. When diffusion loss influences the content of the canonical image, the deformation field is optimized during subsequent training steps. This mutual optimization continues throughout the training process. As a result, by the end of training, our deformation field has already adapted to and is fully compatible with the refined canonical image, requiring no additional adjustments.
>
> > **Q2. LoRA choice**
>
> We chose LoRA for its efficiency and effectiveness in fine-tuning large models with limited data. LoRA's quick convergence and lower computational cost make it particularly suitable for our method. In response to your question, we conducted a comparative experiment with LoHA, another PEFT strategy. Our results show that LoRA achieved a similar visual quality to LoHA in significantly less time (30 minutes vs 70 minutes). This experiment further validated our choice of LoRA for its efficiency.
>
> > **Q3. Large camera movements**
>
> We acknowledge that in scenarios with extremely drastic camera movements or significant changes, our method, like other approaches, does have limitations and may fail to produce ideal results. However, it's important to note that our approach can handle more complex and dynamic scenes compared to existing canonical-based methods. We've included the perceptual results of canonical images in the attached PDF file (Fig. 3(b)) to illustrate this capability.
>
> > **Q4. Multiple canonical images**
>
> We agree this is an important aspect. We have conducted a more in-depth ablation study, the results of which are included in the attached PDF (Fig. 2). Our experiments with varying numbers of canonical images (1, 2, 3, 4, 5) reveal important trade-offs:
> - Too few canonical images: It is challenging to model the entire video, leading to poorer reconstruction quality, especially for complex scenes.
> - Too many canonical images: Decreased temporal consistency and significantly longer training times.
>
> Based on these findings, we empirically use 3 canonical images to strike a balance between editing quality and computational efficiency for most scenarios. This choice allows us to effectively model video content while minimizing the drawbacks associated with extreme numbers of canonical images.
>
> > **Q5. Alternative solutions**
>
> While methods like [A] offer valuable alternatives, our approach of refining canonical images to be natural provides several benefits:
> - Improved editability for downstream tasks.
> - Better preservation of scene semantics.
> - Simplified pipeline without the need for additional combination steps.
>
> To illustrate this, we include visual examples in our attached PDF (Fig. 1) demonstrating why natural canonical images are necessary. These examples show that:
> - Non-natural, distorted canonical images can be challenging to edit in tasks like adding handwritten characters, potentially causing user difficulties.
> - Many current style transfer techniques, such as ControlNet, require natural images as input to function effectively.
>
> > **Q6. Warping error**
>
> Thank you for your valuable suggestions. We implement the long-term warping error metric from [B] as recommended. The table below shows that our method achieves the best performance under this metric, demonstrating the long-term temporal consistency of NaRCan.
>
> |Method|short-term $E_{warp}$[28]$\downarrow$|long-term $E_{warp}$ [B]$\downarrow$|
> |-------------|:--------------:|:--------------:|
> | Hashing-nvd |     0.0070     |     0.0495     |
> | CoDeF       |     0.0085     |     0.0785     |
> | MeDM        |     0.0072     |     0.0583     |
> | Ours        |   **0.0065**   |   **0.0484**   |
>
> Regarding the scale difference with [28], we appreciate you pointing this out. It is due to an oversight in pixel value normalization (0-255 in our evaluations and 0-1 in [28]). We correct this misalignment and re-run our experiments. Our method continues to show the best performance, now with a more consistent scale with [28]. The remaining difference is due to the different tasks: [28] performs blind video temporal consistency enhancement, while NaRCan targets video editing.
>
> We're grateful for your careful review, which has helped us improve our results' accuracy and consistency. These corrections and new evaluations will be included in the updated paper. We're committed to addressing all points raised to improve the final paper.
>
> [28] Lai, Wei-Sheng, et al. "Learning Blind Video Temporal Consistency." ECCV. 2018.
>
> [A] Lei, Chenyang, et al. "Blind Video Deflickering by Neural Filtering with a Flawed Atlas." CVPR. 2023.
>
> [B] Lei, Chenyang, et al. "Blind Video Temporal Consistency via Deep Video Prior." NeurIPS. 2020.

---

> > ### Author Response · Authors · 2024-08-12
> > **Please let us know if you have additional questions after reading our response**
> >
> > Dear Reviewers,
> >
> > We appreciate your reviews and comments. We hope our responses address your concerns. Please let us know if you have further questions after reading our rebuttal.
> >
> > We aim to address all the potential issues during the discussion period.
> >
> > Thank you!
> >
> > Best, Authors

---

> > > ### Comment · Reviewer_b5x9 · 2024-08-13
> > > **The author's response address my concerns.**
> > >
> > > Thanks for your response regarding my concerns. I am glad to see that the author provides a comprehensive explanation and results on Computational efficiency, large camera movements, warping errors, and more discussion on other questions. The response is clear and decent. Hence, I will keep my rating and vote for weak acceptance of this paper.

---

> ### Author Response · Authors · 2024-08-13
>
> Dear Reviewer,
>
> Thank you for your positive feedback on our rebuttal! We're glad our explanations addressed your concerns.
>
> We appreciate your thorough review and remain open to any further questions.
>
> Thank you! Authors

---

### Author Rebuttal · Authors · 2024-08-07

Dear Reviewers and Area Chairs,

We sincerely thank all reviewers for their thorough assessments and valuable feedback. We appreciate the positive comments on our work:

Strengths:
1. Reasonable motivation and significant performance improvements (Reviewer b5x9).
2. Clear experimental analysis (Reviewer b5x9).
3. Superior temporal coherence in video editing results (Reviewer pLuV).
4. Faster convergence compared to existing methods (Reviewer pLuV).
5. Valuable incorporation of homography into our method (Reviewer 4jq5).

We have responded to each reviewer individually to address any comments. We would like to give a brief summary.

1. **Writing clarity and method details**: We expand explanations, particularly for the Hybrid Deformation Field and Diffusion Prior.
2. **Computational efficiency**: We provide specific runtime details for LoRA fine-tuning and the full pipeline.
3. **Justification for design choices**: We explain the selection of LoRA, noise scheduling, and the use of multiple canonical images.
4. **Limitations and failure cases**: We expand the discussion on the types of motions handled, performance on challenging videos, and potential failure scenarios.

We support these with additional experiments on challenging cases, providing both qualitative results in the individual rebuttal below and quantitative results in the attached PDF.

Again, we thank all reviewers and area chairs!

Best, Authors

---

### Decision · Program_Chairs · 2024-09-25

**Decision:**

Accept (poster)

**Comment:**

The paper initially received some mixed reviews, with ratings $(3, 4, 6)$. However, following the rebuttal, all reviewers found the authors' responses adequately addressed their concerns and thus gave the final ratings with two *weak accept* and one *borderline accept*.

Specifically:

- **Reviewer b5x9** found the rebuttal to be clear with a comprehensive explanation and results on computational efficiency, large camera movements, warping errors, and more discussion on other questions.

- **Reviewer 4jq5** confirmed that the proposed method showed better semantic alignment, and the difference between linear interpolation and video frame interpolation mentioned in the paper was clearly explained.

- **Reviewer pLuV** felt the detailed explanation addressed the concerns and believed the remaining obscured parts could be clarified in the revised paper,

Given the positive feedback and support from all reviewers, the area chair recommends accepting this paper.